# New Avenues of Heme Synthesis Regulation

**DOI:** 10.3390/ijms23137467

**Published:** 2022-07-05

**Authors:** Amy E. Medlock, Harry A. Dailey

**Affiliations:** 1Department of Biochemistry and Molecular Biology, University of Georgia, Athens, GA 30602, USA; 2Augusta University/University of Georgia Medical Partnership, University of Georgia, Athens, GA 30602, USA; 3Department of Microbiology, University of Georgia, Athens, GA 30602, USA

**Keywords:** heme, iron, porphyrin, aminolevulinic acid synthase, ferrochelatase, itaconate, posttranslational modification

## Abstract

During erythropoiesis, there is an enormous demand for the synthesis of the essential cofactor of hemoglobin, heme. Heme is synthesized de novo via an eight enzyme-catalyzed pathway within each developing erythroid cell. A large body of data exists to explain the transcriptional regulation of the heme biosynthesis enzymes, but until recently much less was known about alternate forms of regulation that would allow the massive production of heme without depleting cellular metabolites. Herein, we review new studies focused on the regulation of heme synthesis via carbon flux for porphyrin synthesis to post-translations modifications (PTMs) that regulate individual enzymes. These PTMs include cofactor regulation, phosphorylation, succinylation, and glutathionylation. Additionally discussed is the role of the immunometabolite itaconate and its connection to heme synthesis and the anemia of chronic disease. These recent studies provide new avenues to regulate heme synthesis for the treatment of diseases including anemias and porphyrias.

## 1. Introduction

It can be argued that circulating adult mammalian erythrocytes represent one of the most highly evolved and specialized cells in nature. They are not only terminally differentiated, spatially altered, highly specialized cells, but also have undergone enucleation, and loss of functional internal organelles and associated biochemical pathways. This represents the end result of an evolutionary path that started in the annelids where circulating soluble hemoglobin macromolecular complexes were first replaced by specialized hemoglobin-containing cells in the *Capitella* [1]. While circulating red cells continued to evolve within the Metazoa, it is not until mammals that one finds the jump to enucleated cells [2]. Indeed, reptiles, birds, and fish all maintain nucleated circulating red cells. A mini-recapitulation of evolution is seen in developing mammals where the site of erythropoiesis migrates from yolk sac with primitive, nucleated red cells, to fetal liver with fetal red cells, and eventually at birth to bone marrow for adult erythropoiesis and spleen for stress erythropoiesis [3]. Interestingly, with our current knowledge, it is not obvious what critical advantage(s) has been gained by the encapsulation of the soluble, circulating hemoglobin complexes into highly specialized cells. Understanding the complexities of the developmental process that controls erythrocyte production may help answer this question. Some of these answers will only come from the study of mammalian erythropoiesis, since it is only here, and not in any lower animal, where definitive erythropoiesis results in the highly specialized mammalian erythrocyte. Clearly, the identified link between body iron metabolism and red cell development and the sensitivity of this process to infection, demonstrate a strong role for regulatory processes during inflammation and infection whose goal, at least in part, is to sequester iron from invading organisms. Considerable information is now available about many elements of the red cell maturation process and temporal factors that direct it. However, much remains to be learned. Herein, we focus on aspects of the regulation of heme synthesis during definitive erythroid differentiation in mammals.

Erythroid precursors extensively remodel cellular shape and protein content such that one molecule, hemoglobin with its cofactor heme, becomes the single most prevalent protein within the cell. This process, which is initiated by the action of GATA1 starts with hematopoietic stem cells (HSC) where every cell division results in a daughter cell that is distinct from the parent [3,4]. Adding complexity to this is that erythropoiesis occurs within the bone marrow in erythroblastic islands where RBC precursors exist in close association with so-called nurse macrophages (Figure 1) [5,6]. During terminal erythroid differentiation, one of the hallmark events is the coordinated induction of heme and globin synthesis. Mature red cells contain approximately 10^9^ molecules of heme, so a considerable amount of cellular metabolism is devoted to the production of heme and globins during the late stages of differentiation. The onset of heme synthesis occurs when the first pathway enzyme, the erythroid-specific 5-aminolevulinate synthase (ALAS2), is induced. In humans, this occurs in the basophilic erythroblast, reaching a maximum in the polychromatic/orthochromatic erythroblast (Figure 1), while in murine systems the initial induction has been reported to occur slightly earlier in the proerythroblast [7].

All enzymes necessary for catalyzing the individual steps of heme biosynthesis are identical between non-erythroid (housekeeping) and differentiating erythroid cells (Figure 2). Both housekeeping and erythroid-specific transcriptional elements control several genes encoding enzymes of heme synthesis, but two genes exist for the first enzyme, 5-aminolevulinate synthase (ALAS) [8,9,10]. The human gene for the housekeeping 5-aminolevulinate synthase (*ALAS1*) is located on chromosome 3 and is expressed in non-erythroid cells. *ALAS1* expression is subject to regulation by a variety of factors that are beyond the scope of the present review [11]. The *ALAS2* gene is located on the X chromosome and is expressed only in the differentiating erythron. In both non-erythroid and erythroid precursor cells synthesizing heme, it is generally accepted that ALAS activity represents the rate-limiting step of the pathway. GATA1-induced erythroid differentiation results in the upregulation of heme synthesis by induction and enhancement of the transcription of all heme pathway enzyme genes (Figure 3) [11]. Translation of *ALAS2* has been shown to be modulated by iron via the IRE-IRP machinery [12,13,14] and microRNAs [15]. Posttranslational hydroxylation decreases the activity of ALAS2 and destabilizes it for proteasomal degradation [16], and post-translational translocation of the protein into the mitochondrion is inhibited by the binding of heme to the cytoplasmic precursor form of the enzyme [17,18,19].

A substantial body of data exists on mechanistic features of individual enzymes [11,20] and transcriptional regulation of the heme biosynthetic pathway during erythroid differentiation in animals, erythroleukemia (both murine erythroleukemia (MEL) and human erythroleukemia K562), embryonic stem [21,22], and CD34+ cell models [4,7]. Additionally, data now exist that characterize mechanisms involved in the supply of iron [23] and its role in the regulation of heme synthesis during erythroid differentiation, but much less attention has been paid to the supply of the organic precursors’ glycine [24,25] and succinyl-CoA [26] for the synthesis of the tetrapyrrole macrocycle. Recent studies also make clear that additional regulatory features may exist in situ as protein–protein interactions occur in the multiprotein complex named the mitochondrial heme metabolon [27,28,29]. However, little is known about the potential regulation of heme biosynthesis enzymes via post-translational modification (PTM) even though PTMs such as phosphorylation, acylation, and glutathionylation are well-acknowledged regulatory mechanisms in metabolic pathways across many cellular compartments [30,31,32]. Monitoring and understanding regulatory schemes are challenging, but essential if one is to understand the metabolic basis for diseases. Herein, we review recent work related to the provision of glycine and succinyl-CoA for heme production during erythropoiesis and highlight the possible roles of PTM of heme synthetic enzymes as regulators of tetrapyrrole synthesis.

## 2. Provision of Glycine and Succinyl-CoA for 5-Aminolevulinic Acid (ALA) Synthesis during Erythropoiesis

Protoheme IX has a total of 34 carbons, with 26 of these derived from succinyl-CoA and 8 from glycine (Figure 4). The synthesis of each molecule of protoporphyrin IX requires eight molecules of glycine and eight molecules of succinyl-CoA to form the tetrapyrrole macrocycle. In nonerythroid cells where modest amounts of heme are synthesized, the sources for these compounds may be from housekeeping intermediary metabolism associated with the tricarboxylic acid (TCA) cycle and serine metabolism. Homedan et al. [33] demonstrated that hepatic mitochondria treatments, which have a negative impact on the TCA cycle, cause diminished heme synthesis. One caveat to this conclusion is that they noted a significant decrease in α-ketoglutarate dehydrogenase (KDH), which, by itself, would limit succinyl-CoA synthesis. In developing erythroid cells where large amounts of heme are synthesized in a relatively short period of time, obtaining glycine from serine metabolism and succinyl-CoA from the TCA cycle would require a significant replenishment via anaplerotic reactions.

Regarding the source of glycine for heme synthesis during erythropoiesis, there is good evidence that plasma glycine, which is present at around 250 micromolar concentrations, is the main source. Data from a number of groups demonstrate that the plasma membrane glycine transporter GlyT1 and mitochondrial transporter SLC25A38 are essential for normal red cell development [24,25,34,35]. A deficiency in either transporter has a negative impact on the synthesis of heme and results in anemia. Specifically, mutations in the SLC25A38 gene are reported to be the cause of inherited recessive sideroblastic anemia [24,36]. With regard to the essential nature of plasma membrane glycine transport, it has been shown that GlyT1^−/−^ mice develop hypochromic microcytic anemia, which is lethal at one-day postnatal [25]. Chemical inhibition of GlyT1 in rats induced a steady-state microcytic hypochromic anemia that is pathogenetically distinct from systemic iron-overload anemias [37]. This suggests that either cellular glycine synthesis is insufficient for tetrapyrrole synthesis or that the cellular pool of glycine produced within developing red cells is not available to ALAS2.

How succinyl-CoA is provided for heme synthesis during erythroid differentiation was not well understood nor researched until relatively recently [26]. Assuming that glycine for heme synthesis is exogenously supplied [25,35], then 76% of the heme carbons originate from an internal metabolic source that produces succinyl-CoA (Figure 4). The long-accepted assumption was that the TCA cycle supplies succinyl-CoA for RBC heme synthesis. Data presented in support of this proposal was the observation that ALAS2, but not ALAS1, physically interacts with the ATP-dependent succinyl-CoA synthetase (SCS) β-subunit (SUCLA2) [38,39]. It was proposed that since ALAS2 utilizes succinyl-CoA to synthesize ALA, and because SUCLA2 is a subunit of SCS involved in the ATP-dependent reverse reaction of SCS to generate succinyl-CoA from succinate, the ALAS2–SUCLA2 interaction exists to provide succinyl-CoA for ALA synthesis during erythropoiesis. However, while it is clear that an association between ALAS2 and SUCLA2 can exist, there are no data available to demonstrate what role, if any, this interaction plays in the supply of succinyl-CoA for ALAS2. Interestingly, the original publication that reported the ALAS2 and SUCLA2 interaction proposed three possible explanations for the interaction with only one of these being involved in the supply of succinyl-CoA for ALA synthesis [38]. 

Several additional factors argue against SUCLA2-dependent reverse SCS as a supplier of succinyl-CoA for ALA synthesis. First, early studies of particles prepared from fractionated anemic chicken reticulocytes showed that α-ketoglutarate, when compared to succinate, served as a better precursor for ALA synthesis [40,41]. Second, from an overall carbon economy perspective, using intermediates of the TCA cycle for ALA production would deplete all TCA cycle intermediates and require large-scale replenishment via anaplerotic reactions. In addition, the reverse SCS reaction to generate succinyl-CoA from succinate would involve a significant energetic cost to the cell. Third, mice heterozygous for *SUCLA2* knockout [42] and human patients with SUCLA2 deficiency (MIM ID#612073) are not anemic as would be expected if the ALAS2 and SUCLA2 interaction are essential for heme synthesis. Fourth and related, ALAS2 variants which display decreased interactions with SUCLA2 do not have a uniform impact on ALAS2 activity. Some of these variants result in increased ALAS2 activity causing X-linked protoporphyria [43] and other variants have diminished ALAS2 activity causing sideroblastic anemia [38,39].

The role that SUCLA2 plays is currently unknown. It binds to both ALAS2 and FECH [27] so it can be imagined that during the early stages of erythroid differentiation it may interact with and stabilizes the apoprotein forms of ALAS2, which requires pyridoxal phosphate as a cofactor [44], and FECH, which requires the presence of a [2Fe-2S] cluster [45]. An alternate possibility is that by binding to ALAS2 and FECH, SUCLA2 may prevent assembly of the ATP-dependent SCS and, thereby, reduce the conversion of succinyl-CoA into succinate by SCS when ALAS2 levels are high. Of note, there is no stimulation of ALAS2 activity in the presence of SUCLA2.

As mentioned above, if the TCA cycle were to function in situ to supply succinyl-CoA for ALA synthesis, TCA cycle intermediates would require replenishment via anaplerotic supplementation [46]. This replenishment of the TCA cycle could occur from glucose by the conversion of pyruvate into oxaloacetate and glutamine via the conversion of glutamate into α-ketoglutarate. Additionally, it would be anticipated that enzymes essential for the TCA cycle to function would be elevated during erythropoiesis in order to keep up with cellular demands for TCA cycle intermediates, which are needed for not only heme synthesis, but also amino acid biosynthesis to synthesize globin molecules. However, examination of data in ErythronDB [47,48] for cells undergoing definitive adult erythropoiesis reveals that while mRNA levels for the *KDH subunit E1* have a positive correlation with heme synthesis enzymes *ALAS2* (0.92) and *FECH* (0.90) (which both show significant increases during erythropoiesis) there are negative correlations with the mRNA for TCA cycle enzymes *succinate dehydrogenase complex iron-sulfur subunit B (SDHB)* (−0.49), *SUCLA2* (−0.55), *ATP Citrate Lyase (ACLY)* (−0.57), and *succinate dehydrogenase complex subunit D* (*SDHD)* (−0.75) which generally remain constant or decrease. Additionally, we demonstrated that among TCA cycles enzymes only three, KDH, aconitase, and isocitrate dehydrogenase, showed an increase in enzyme activity and the remainder decreased [26]. Interestingly KDH activity, protein, and mRNA levels all increased to a much greater extent than other TCA cycle enzymes. 

KDH, which was previously known to be a nucleating point for the TCA cycle metabolon [49], was also found to interact strongly with ALAS2. When purified, His-tagged, recombinant ALAS2 and untagged KDH purified from porcine hearts were mixed in solution and applied to a cobalt resin column, they are co-eluted. Additionally, when in situ interactions between KDH and ALAS2 were probed with antibody pulldowns of mitochondria from differentiating MEL cells using FLAG-tagged ALAS2, ALAS2 was found to interact with the E1, E2, and E3 subunits of KDH [26]. Thus, both in vitro and in situ approaches support interaction between ALAS2 and KDH.

Burch et al. [26] examined the source of succinyl-CoA for heme synthesis during erythropoietic differentiation of MEL cells by employing ^13^C labeled glucose, diethyl succinate (DES), and glutamine; all possible cellular precursors to succinyl-CoA. Quantitation of label incorporated into succinyl-CoA along with isotopomeric data clearly demonstrated that glutamine is by far the preferred source of carbon for succinyl-CoA synthesis over glucose and succinate. Additionally, the isotopomer data indicated that glutamine was utilized directly without passage around the TCA cycle. Interestingly, in differentiating MEL cells employed in that study, glucose is converted into the TCA cycle intermediates fumarate and malate more efficiently than glutamine and cellular glutamate pools are enriched with label from glucose, suggesting that glucose may make contributions to heme via the cellular glutamate pool rather than via the TCA cycle. This is consistent with glutamine that is preferentially converted into succinyl-CoA for use by ALAS2 and not for a generic replenishing of the TCA cycle.

To further probe the nature of the precursor(s) of α-ketoglutarate and succinyl-CoA for heme synthesis, aminooxy-acetic acid (AOA), a broad-spectrum inhibitor of PLP-dependent transaminases that target enzymes potentially required for the synthesis of succinyl-CoA, was examined in cells undergoing erythropoiesis. AOA was known to block early erythroid differentiation of CD34+ HSC [50] by blocking glutamine deamination and, thus, the eventual conversion of glutamine into α-ketoglutarate [51]. A bypass of this block can be achieved by the addition of dimethyl-α-ketoglutarate (DMK), a cell-permeable analog of α-ketoglutarate. The addition of AOA at the initiation of erythroid differentiation of MEL cells reduced hemoglobin content by 76% [26]. Hemoglobinization was rescued by the addition of either DMK or ALA, but not DES, to AOA-treated differentiating MEL cell cultures. These data are consistent with the succinyl-CoA for heme synthesis during erythropoiesis arising from α-ketoglutarate, and not succinate from the TCA cycle. Similarly, in cultured human CD34+ cells undergoing erythropoiesis and beginning to hemoglobinize, AOA inhibits hemoglobinization, and this inhibition is rescued by the addition of DMK and ALA, but not DES or glucose. 

These data indicate that a major role served by glutamine during hemoglobinization must be for heme synthesis since ALA alone overcame the AOA block during the late stages of erythropoiesis. Consistent with the proposal that ALA is derived from glutamine via KDH rather than from succinate via SCS is the clinical observation that deficiency in thiamine, a required cofactor for KDH, results in sideroblastic anemia [52,53] as found in X-linked sideroblastic anemia caused by a deficiency in ALAS2 activity. Overall, published data suggest that in developing erythroid cells, exogenously supplied glutamine is the source of succinyl-CoA via KDH operating independently of the TCA cycle in a moonlighting role (Figure 5).

## 3. Role of the Immunometabolite Itaconate on Erythropoiesis

One mechanism for the development of anemia is dysfunctional heme synthesis. The two most common anemias are anemia of chronic disease/inflammation (ACD) and iron deficiency anemia (IDA). A significant body of research currently exists related to iron metabolism for heme synthesis; thus, considerable information exists concerning the basis of IDA. On the other hand, the pathophysiology of ACD is less clear. If ACD were caused by sequestration of iron alone, then one would expect to see similar cellular phenotypes with both anemias, but there are significant differences. IDA is characterized as hypochromic microcytic anemia, whereas ACD is usually normochromic normocytic anemia [54]. This implies that factors other than simply iron availability for erythropoiesis must be in play in ACD.

It is known that significant alterations occur in the erythropoietic program during inflammation that are mediated via macrophages. These macrophages, generally referred to as nurse macrophages, maintain close proximity to developing erythroid cells in erythroblastic islands and are known to provide nutrients (Figure 1) [55,56]. One model for the study of ACD is the production of an inflammatory response by treatment of animals or cultured cells with lipopolysaccharide (LPS). LPS-mediated inflammation in mice results in anemia via inhibition of erythropoiesis in bone marrow at an early stage, with an observed depletion of proerythroblasts and basophilic erythroblasts [57], the stages during which ALAS2 is first induced [7]. 

During inflammation *Irg1* (immunoresponsive gene 1) is one of the most highly up-regulated genes in macrophages under pro-inflammatory conditions [58,59,60,61,62,63]. *IRG1* encodes aconitate decarboxylase (IRG1, also called ACOD1) which catalyzes the production of itaconate from the TCA cycle intermediate aconitate [62]. Itaconate is a compound that has antimicrobial properties and when accumulated intracellularly disrupts the TCA cycle by blocking succinate dehydrogenase resulting in increased intra- and extracellular succinate [59,60,61,63]. Activated macrophages produce up to 10 mM itaconate and excrete it into the surrounding milieu, which in situ includes erythroid precursor cells of the erythroblastic island [64]. Itaconate is not synthesized by developing erythroid cells, but it is actively transported into these cells [65], possibly by SLC13A3 a promiscuous plasma membrane transporter [66]. Since itaconate causes accumulation of intracellular succinate that could be converted into succinyl-CoA by SCS, it would seem that heme synthesis might increase on exposure to millimolar concentrations of itaconate. However, this is not the case [64]. When itaconate is added to cultures of differentiating cells either as a neat compound or as a medium from LPS-stimulated macrophages, there is decreased hemoglobinization [67].

By employing ^13^C labeled itaconate, it was found that intracellular itaconate is converted into ^13^C labeled itaconyl-CoA [65]. The source of this itaconyl-CoA is not via SCS as was long proposed [68,69], but has been shown to be formed by the mitochondrial type III CoA acyl transferase named succinyl CoA: glutaryl CoA-transferase (SUGCT) [65]. SUGCT was shown to exchange the succinate of succinyl-CoA with itaconate to form itaconyl-CoA and succinate. Exogenously supplied itaconate can be transported into the mitochondrion where SUGCT resides by the mitochondrial transporters SLC25A10 (succinate transporter) and SLC25A1 (citrate transporter) [70]. The mechanism by which itaconate diminishes heme synthesis is by inhibition of ALA formation. Of note is that SUGCT not only produces itaconyl-CoA, but it also consumes succinyl-CoA in the process and, thereby, reduces the normal ALAS2 substrate succinyl-CoA as it forms itaconyl-CoA. However, this is not its sole avenue of action. Itaconate has no impact on ALAS2 activity and itaconyl-CoA is not a substrate of ALAS2, but it is a competitive inhibitor with a *K*_i_ of approximately 100 μM [65]. Itaconyl-CoA is also a competitive inhibitor of KDH. Thus, itaconyl-CoA inhibits hemoglobinization by directly inhibiting ALAS2 as well as diminishing the level of cellular succinyl-CoA via the SUGCT transferase reaction and by inhibition of KDH (Figure 6).

The action of itaconate on heme synthesis resembles the action of hepcidin on heme synthesis during chronic infection [71]. Hepcidin and itaconate are antimicrobial compounds produced during the inflammatory response. Both inhibit heme synthesis distinctly: hepcidin by regulating iron and itaconate by regulating porphyrin synthesis. Thus, these represent complementary systems that prevent the accumulation of protoporphyrin and iron, two toxic compounds, and regulate overall heme synthesis during inflammation.

## 4. PTMs of Heme Synthesis Enzymes

Reversible PTMs of proteins are recognized as critical regulators of cellular metabolism, but only modest attention has been paid to post-translational regulation of enzyme activity related to heme synthesis enzymes. Studies over the past decade have identified phosphorylation, lysine acylation, and cysteine glutathionylation as key components of regulatory circuits in a variety of metabolic pathways and demonstrated that disruption of normal PTM can result in disease states [72,73,74]. Interestingly, recent data on mitochondrial PTMs have established a link between succinylation and glutathionylation via NADPH levels showing that both of these forms of PTM may be subject to redox regulation in the mitochondrion [75]. Since heme is a key player in redox chemistry, factors that have an impact on the cellular redox state might be anticipated to play a role in regulating heme biosynthesis. PTMs can be irreversible or reversible, thus enhancing the dynamics of cell signaling and homeostasis. Though not always considered a PTM, cofactor assembly can be included and is relevant to heme synthesis and redox regulation. To date, only a few published studies exist for PTMs involved in the regulation of heme synthesis enzymes. These are the reversible glutathionylation of C52 in uroporphyrinogen decarboxylase in *Saccharomyces cerevisiae* [76], the phosphorylation of FECH at threonine 116 during erythropoiesis [77], and the pH regulation of the FECH [2Fe-2S] cofactor [78]. In addition to modulating enzyme activity, these PTMs could impact the formation of protein–protein interactions of the mitochondrial heme metabolon [27,29] and control substrate flux as well as the product release.

### 4.1. Cofactor Assembly

Five heme synthesis pathway enzymes have been shown to possess cofactors that are assembled post-translationally. These are a dipyrromethane in hydroxymethylbilane synthase (HMBS, previously known as porphobilinogen deaminase, PBGD) which is self-assembled in the first catalytic turnover [79,80], a pyridoxal phosphate in both ALAS1 and ALAS2 [81,82], a recently identified [4Fe-4S] cluster in porphobilinogen synthase [83], a flavin adenine dinucleotide (FAD) cofactor in PPOX [84], and a [2Fe-2S] cluster in FECH [45]. Currently, data for the potential of any of these cofactors to serve a post-assembly regulatory function are only available for FECH. Mammalian FECHs have a [2Fe-2S] cluster that is required for activity [45]. This [2Fe-2S] cluster functions as a pH and membrane potential-sensitive regulator of FECH activity [76], thus connecting heme synthesis with the redox state of the cell. Because of the role of the cluster as a pH and membrane potential sensor, any structural alterations that have an impact on the cluster’s midpoint potential may have an impact on in vivo activity.

### 4.2. Phosphorylation

With regard to phosphorylation of FECH at T116, enzyme kinetic analysis revealed an increase in enzyme activity [77], suggesting that this PTM is significant to alter catalysis. Since T116 is highly conserved in eukaryotic FECHs, this PTM could be an indispensable process for metazoans during hemoglobinization. To buttress that suggestion, a knock-in of T116A to eliminate the site of phosphorylation site resulted in decreased FECH activity in murine erythroleukemia cells [77].

FECH is targeted in vivo for phosphorylation by protein kinase C (PKC) and/or protein kinase A (PKA) [77,85]. Phosphorylation by PKA may be more physiologically relevant than PKC, as PKC-mediated FECH phosphorylation did not result in a change in the enzyme’s activity [85]. In contrast to PKC, PKA is upregulated during erythropoiesis [77,86,87]. Interestingly PKA is localized to the outer mitochondrial membrane via interactions with an anchoring protein, AKAP10 [77]. Thus, phosphorylation of FECH at T116 must occur post-translationally, but prior to import into the mitochondrion and assembly of the [2Fe-2S] cluster. The T116 residue is in the middle of a long α-helix that we have shown moves about 2 Å upon porphyrin binding [88]. T116 is within hydrogen-bonding distance with H86, which is located on an adjacent helix, and Q120, which is located on the same helix (Figure 7). Phosphorylation of T116 introduces a larger, charged functional group that is predicted by in silico modeling to cause the helix to be shifted a few Å towards the active site pocket opening and/or introduce a kink in the helix. Either of these structural alterations would be expected to have an impact on the molecular reorganization that occurs during catalysis. Previously, we have demonstrated with steady-state and stopped-flow analysis that product (heme) release is the rate-limiting step by an order of magnitude for in vitro catalysis of FECH [89]. Thus, it might be anticipated that a PTM causing an increase in activity may do so by increasing the rate of product release. Kinetic analysis of the partially phosphorylated FECH (~10%), showed an increase in the V_max_, but not the K_m_ relative to the unphosphorylated form [77]. Preliminary experiments in our laboratory have also identified additional sites for phosphorylation of FECH. The physiological significance of these PTMs is currently under investigation. ALAS2 lacks canonical threonine or serine phosphorylation motifs and there is no evidence currently available to suggest that it is sensitive to PTM by phosphorylation.

### 4.3. Glutathionylation

Cysteine residues within proteins are frequently buried, but when found on the surface act as molecular redox switches which can be important for protein structure and regulation [90,91]. The thiol group of cysteines can undergo several modifications when converted to the more reactive thiolate form. Glutathionylation of cysteines is a redox-dependent process that involves the attachment of glutathione (GSH) and is one of the most common modifications of this residue [72,92]. As mentioned above, a single study on glutathionylation of uroporphyrinogen decarboxylase demonstrated that only one of three cysteine residues, C52, was modified and that this was reversible by Grx2p [76]. There were no enzymatic assays performed but given the closeness of C52 to the active site, it may be expected that this PTM would impact enzyme activity.

Evaluation of the potential for glutathionylation of the four mitochondrially located heme synthetic enzymes, ALAS2, FECH, coproporphyrinogen oxidase (CPOX), and PPOX, along with one cytoplasmically located enzyme, HMBS was performed using the biotinylated glutathione ethyl ester (BioGEE) system in western blots [93,94]. The data reveal that ALAS2, FECH, and CPOX, but not PPOX or HMBS, are subject in vitro to reversible glutathionylation (Figure 8). When in vitro glutathionylated ALAS2 and FECH were subjected to nano LC-MS/MS detection and MASCOT-based bioinformatics analysis, residues C195 and C395 of ALAS2, and C323 and C395 of FECH were found to be glutathionylated. Although there are a total of seven conserved C residues in ALAS2s, only C195 and C395 were identified as being glutathionylated. These two residues are located near the protein surface and are not within the active site pocket. For FECH, C323 and C395 represent two of the three conserved C residues aside from the four involved in the [2Fe-2S] cluster [95]. C323 is surface exposed, and in some of our crystal structures, it forms a disulfide bond with C360, a residue that is not conserved in zebrafish FECH. C395 is located adjacent to the solvent-filled tunnel that contains the [2Fe-2S] cluster [95]. Our structure-function studies of FECH demonstrated that the tunnel in which the [2Fe-2S] cluster resides is sufficiently open and solvent accessible to allow for the passage of molecules as large as salicylic acid [96,97]. Given the proximity of C395 to the cluster, its glutathionylation could have a significant impact on the cluster midpoint potential and possibly FECH enzyme activity. This may have physiological consequences, since our previous studies linked FECH activity in vivo, via its [2Fe-2S] cluster, to mitochondrial pH and redox state [78].

When FECH activity is measured in vitro in the presence of 3mM GSH, its activity is significantly enhanced (Figure 9a). When recombinant FECH is incubated in vitro with Glrx 5 +/− GSH and the modified FECH is then assayed in a buffer lacking GSH, the Glrx5-mediated PTM of FECH causes a 40% increase in activity when compared to unmodified WT FECH (Figure 9b). However, the activity of a FECH C395S variant was only modestly stimulated by GSH, whereas the C323V variant had near wild-type stimulation by GSH (Figure 9a). This suggests that the tunnel-located C395, but not C325, may be significant for the regulation of FECH activity by PTM.

### 4.4. Succinylation

Multiple investigators have identified and reported MS-generated libraries of protein succinylation sites for human, mouse, and microbial cells [74,101,102]. Among all databases, one finds FECH succinylated at multiple lysine residues. Since erythroid cells were not included in any of the analyses, there are no data for succinylation of ALAS2. The succinylation of lysyl residues is a spontaneous chemical process with succinyl-CoA, so it is not surprising to find proteins within the mitochondrial matrix that are succinylated since succinyl-CoA is present there and essential for heme biosynthesis. The reversal of succinylation is enzymatically mediated by the mitochondrially located SIRT5 [103,104].

Succinylation of ALAS2 in vitro results in a 30–50% increase in enzyme activity. The increase in activity of ALAS2 via reversible succinylation adds a new level to the regulation of heme synthesis via a feed-forward activation mechanism. While feed-forward activation by iron has been proposed via the [2Fe-2S] cluster for FECH, such regulation for ALAS2 has not been suggested. This novel mechanism of ALAS2 may result in the identification of mutations that result in idiopathic sideroblastic anemias or iron refractory anemias.

Although multiple studies have identified succinylation of three conserved lysine residues (human numbering K138, K290, and K415) of FECH [74,101,102], there have not been any reported changes in the activity of FECH for the succinylated vs. unmodified enzyme. However, this does not necessarily mean that the succinylation of FECH has no in vivo consequences. Succinylation may have an impact in situ on the ability and/or efficiency of FECH to interact with other proteins in the heme metabolon [27,29]. This suggestion is supported by the observation that molecular recontouring, shown to occur during the catalytic cycle [88,105], results in alterations in the ability of FECH to interact with other proteins [29]. Additionally, we have found that purified ALAS2 and FECH form a complex in vitro [27] and that in this complex, FECH enzyme activity is stimulated 40%. Alternatively, or additionally, succinylation may affect in vivo activity by altering the midpoint potential of the FECH [2Fe-2S] cluster since K415 is located adjacent to the cluster in the solvent-filled channel. We have previously demonstrated the importance of this cluster and its potential regulatory role in anemic zebrafish [78]. Interestingly K415, which is effectively desuccinylated by SIRT5 in vitro (Figure 10), is located spatially close to C395, a residue identified (above) as being subject to glutathionylation and located within the solvent-filled channel by the [2Fe-2S] cluster [95]. The presence of two different PTM susceptible residues (C395 via glutathionylation and K415 via succinylation) within the same region and in close vicinity of the required cofactor (Figure 11) is intriguing and deserving of particular focus.

## 5. Conclusions

Recent studies described herein provide new avenues to further investigate the regulation of heme synthesis in order to understand the fine-tuning of the synthesis of this essential cofactor. While the synthesis of heme for the development of erythroid cells has been the focus of most studies, these findings are applicable to other cells where the demand for heme can change rapidly to accommodate processes such as steroidogenesis and xenobiotic detoxification. These new areas to understand heme synthesis will help us better understand normal developmental and cellular processes as well as pathological conditions, including porphyrias and anemias [20,54,106]. As noted throughout, there are clear connections to precursor availability, post-translational modifications and inflammation to heme synthesis in these pathological conditions [14,24,25,34,52,53,57,77]. For these pathological states, there is a desperate need for new treatment strategies that can be uncovered by a better understanding of the endogenous regulation of heme biosynthesis.

## Figures and Tables

**Figure 1 ijms-23-07467-f001:**
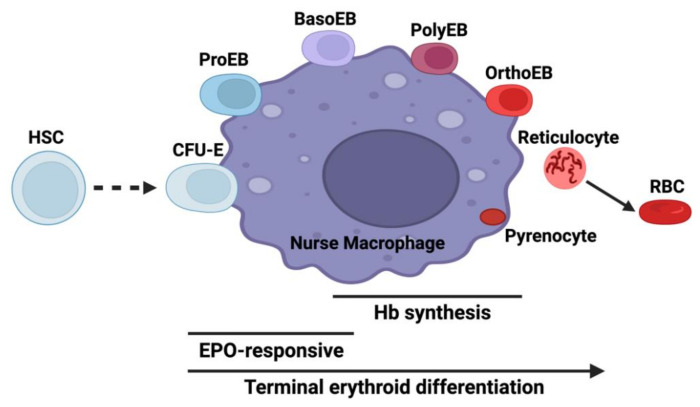
Erythroid differentiation in the context of the erythroblastic island niche. The diagram illustrates interactions between developing erythroid cells and the nurse macrophage. Abbreviations are HSC: hematopoietic stem cell, CFU-E: colony-forming unit-erythroid, ProEB: proerythroblast, BasoEB: basophilic erythroblasts, PolyEB: polychromatic erythroblast, OrthoEB: orthochromatic erythroblast, RBC: red blood cell, Hb: hemoglobin, and EPO: erythropoietin. Created with BioRender.com.

**Figure 2 ijms-23-07467-f002:**
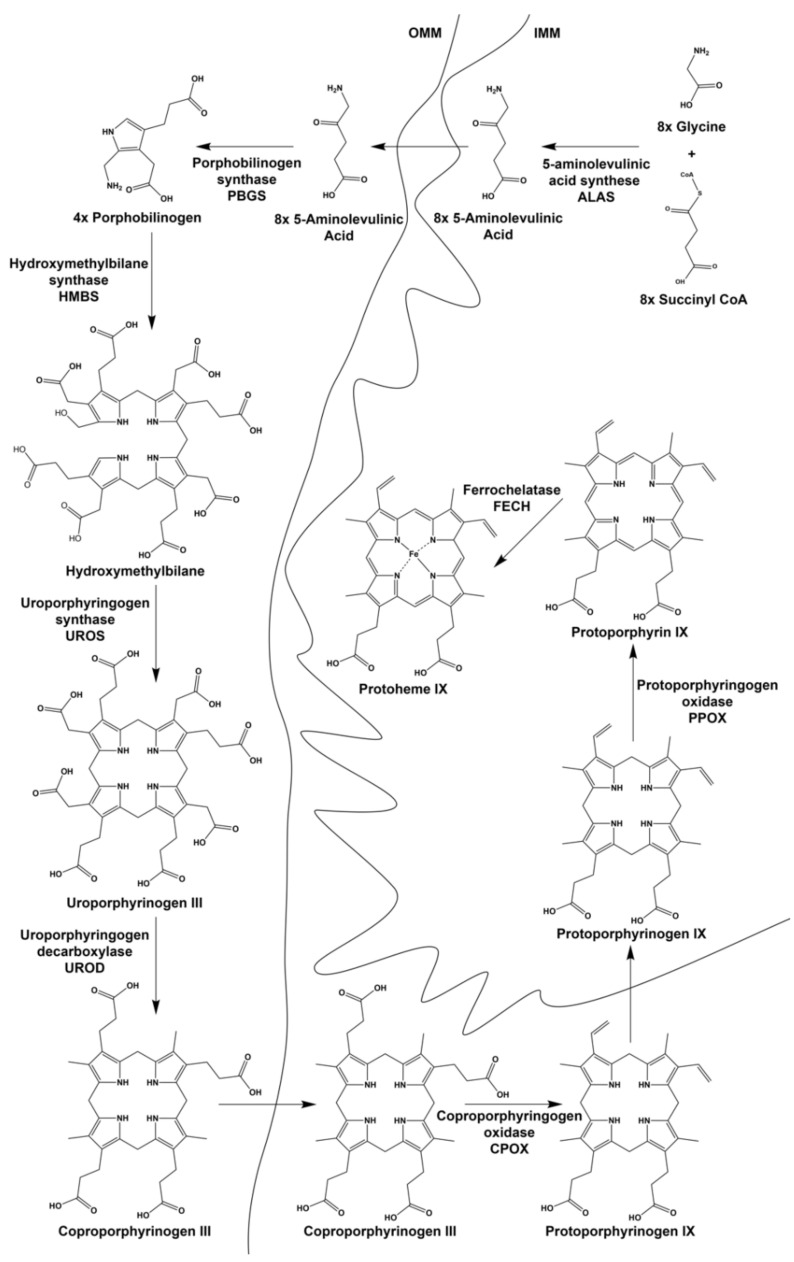
Intermediates and enzymes of the heme biosynthetic pathway. Both intermediates and enzymes are shown as well as their cellular localization. OMM is the outer mitochondrial membrane and IMM is the inner mitochondrial membrane.

**Figure 3 ijms-23-07467-f003:**
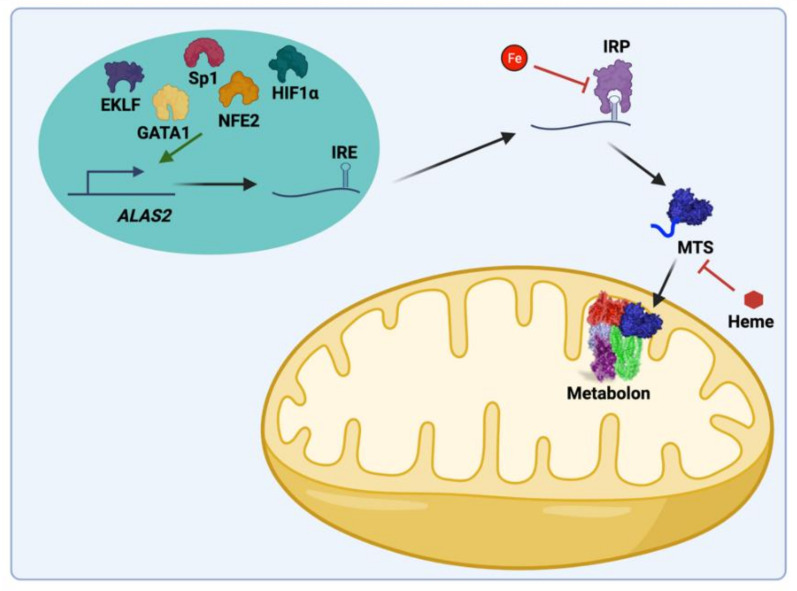
Key regulatory features of mammalian ALAS-2 synthesis, stability, and activity. *ALAS2* transcription is regulated by GATA1, NfE2, EKLF, Sp1, and HIF1α. *ALAS2* mRNA possesses an iron-responsive element (IRE) that is bound by iron-responsive protein (IRP) in the absence of iron inhibiting translation. Additionally, heme can regulate the import into the mitochondria. In the mitochondria, the ALAS2 protein is regulated by the mitochondrial heme metabolon, which includes ALAS2, FECH, PPOX, SCS, and KDH. Created with BioRender.com.

**Figure 4 ijms-23-07467-f004:**
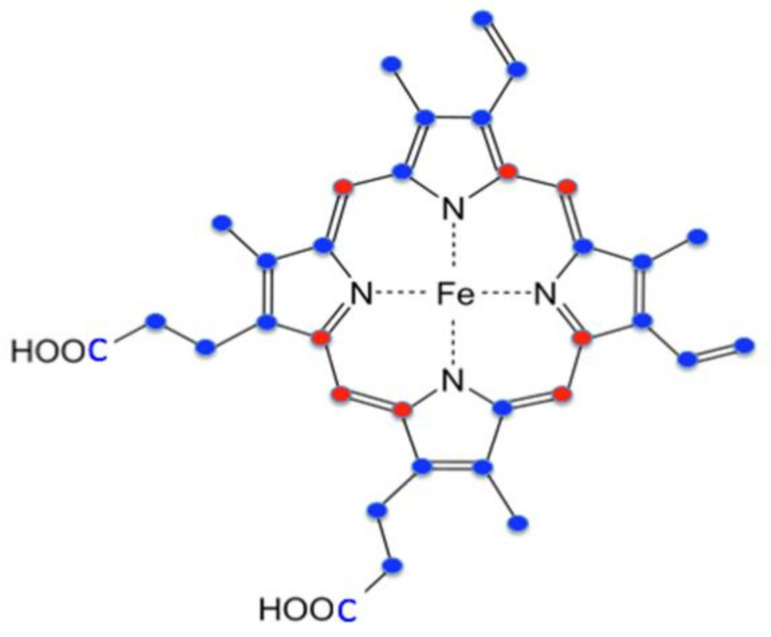
Origin of heme carbon atoms. Red-colored carbons are derived from glycine. Blue-colored carbons are derived from succinyl CoA.

**Figure 5 ijms-23-07467-f005:**
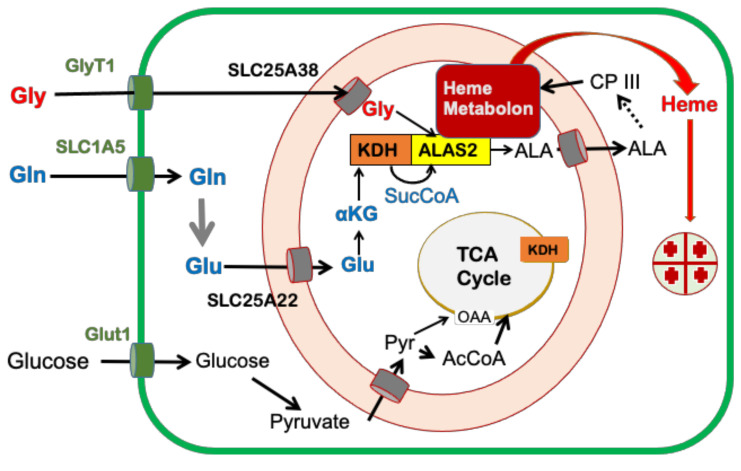
Cellular metabolic pathways that contribute to mammalian heme synthesis during erythropoiesis. The cartoon adapted from [26] shows the origin of carbon for heme synthesis. The plasma membrane is shown in green and the mitochondrial inner and outer membranes are represented in orange.

**Figure 6 ijms-23-07467-f006:**
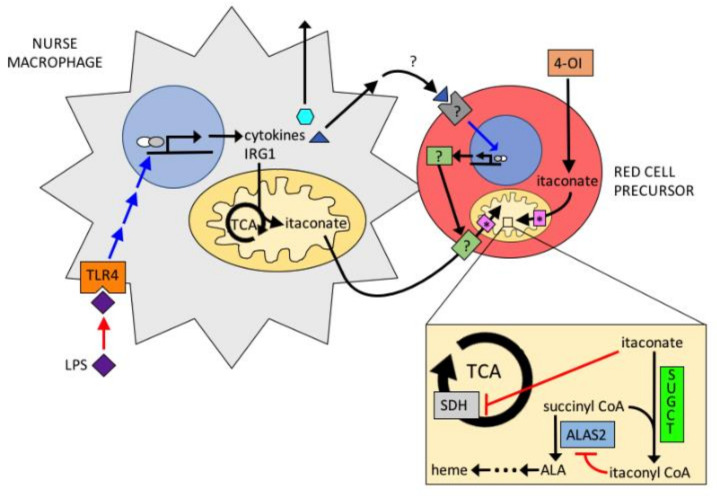
Proposed model for the action of itaconate in erythropoiesis. Itaconate is produced by nurse macrophages via LPS stimulation. Itaconate is imported into erythroid precursors and converted to itaconyl-CoA by SUGCT. Itaconyl-CoA inhibits ALAS2 and KDH.

**Figure 7 ijms-23-07467-f007:**
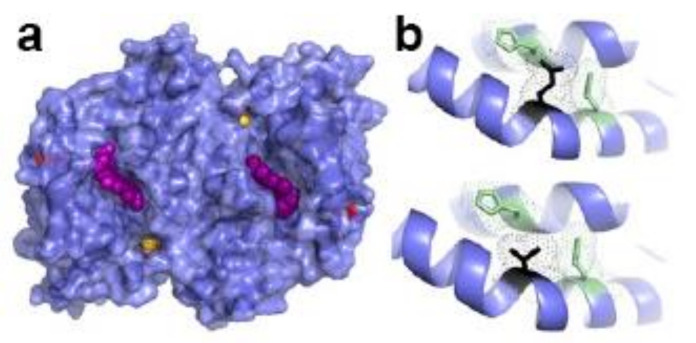
Structure of WT human FECH. (**a**) Surface representation of FECH dimer. Heme shown as purple spheres, [2Fe-2S] clusters yellow and orange spheres, and T116 as red surface. (**b**) Bottom, residues within hydrogen-bonding distance to T116 (black) include H86 and Q120, (green). Top, mutation of T116E (black), which would resemble phosphorylated T116 in size and charge is physically too large to fit in the unphosphorylated structure and would, therefore, alter orientation and hydrogen bonding between the two helices shown. Figure created using PyMol (Schrödinger, Inc., New York, NY, USA) and PDB ID 2HRC.

**Figure 8 ijms-23-07467-f008:**
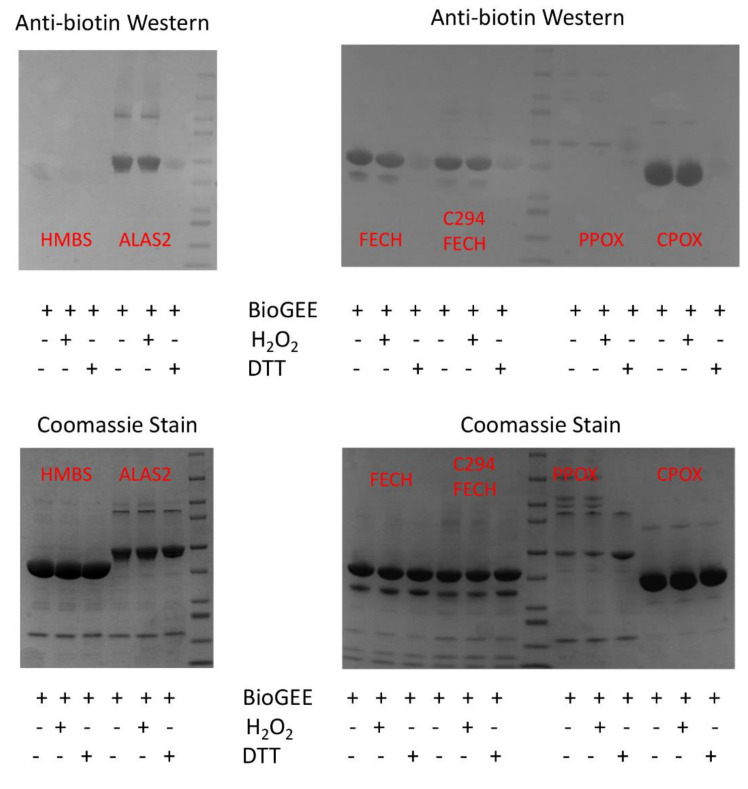
Some recombinant human heme biosynthetic enzymes are glutathionylated by BioGEE. Medlock and Dailey personal communication. Purified recombinant human heme biosynthetic enzymes [81,84,98,99], approximately 100 μM, were incubated with 500 μM BioGEE. In some samples, 1 μM H_2_O_2_ was added to facilitate glutathionylation via a sulfenic acid intermediate, and in some 10 mM DTT was added to demonstrate thiol specificity. Samples were separated by SDS-PAGE and subjected to Western blots using anti-biotin (Sigma) (top) and Coomassie staining (bottom). A representative of the two experiments is shown.

**Figure 9 ijms-23-07467-f009:**
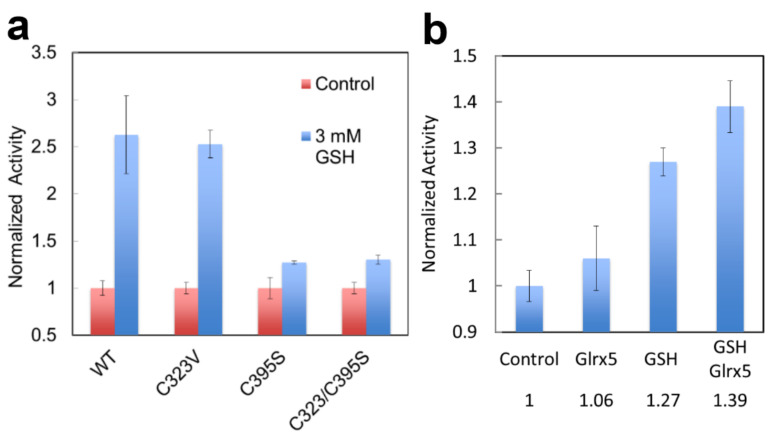
Enzyme activity of glutathionylated human FECH. Medlock and Dailey personal communication. (**a**) The activity of WT and C variants assayed +/− 3 mM GSH. (**b**) The activity of FECH pre-incubated with Glrx5, 50 mM GSH, or both, and then assayed in the absence of GSH. Proteins were purified [98] and assayed [100] as previously described. Experiments were performed in triplicate.

**Figure 10 ijms-23-07467-f010:**
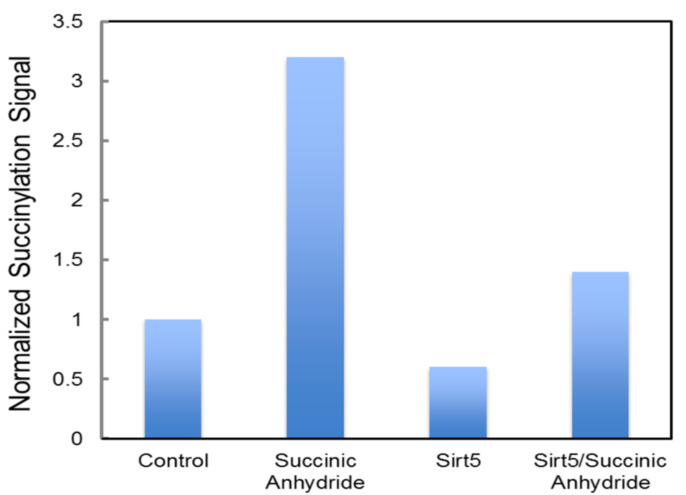
Desuccinylation of human FECH by SIRT5. Medlock and Dailey personal communication. Recombinant FECH, ~100 μM, was incubated with a 50-fold excess of succinic anhydride to succinylated FECH. Isolated, unsuccinylated FECH, and succinylated FECH samples were exposed to purified SIRT5. Quantitation of succinylation was via Western blot density analysis following SDS-PAGE. Anti-succinyl-lysine (PTM Biolabs) was used to detect succinylated protein. Proteins were purified as previously described [98].

**Figure 11 ijms-23-07467-f011:**
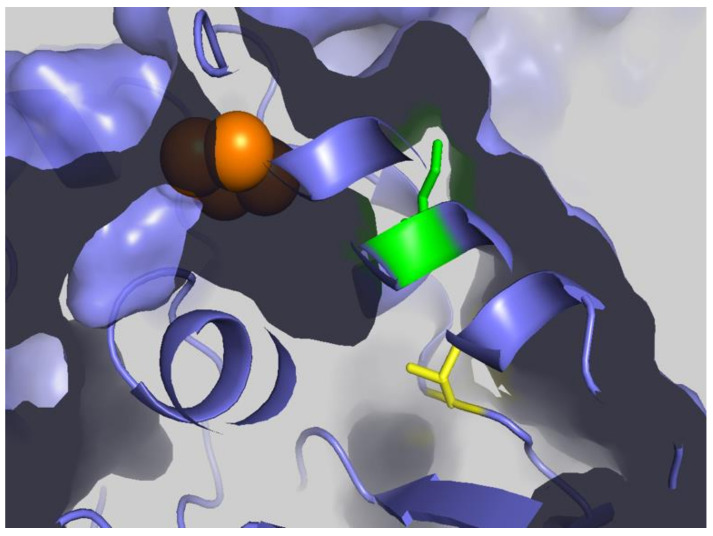
FECH Cluster tunnel. Surface representation of FECH with [2Fe-2S] clusters shown as orange spheres, K415 shown as green, and C395 shown as yellow sticks. Figure created using PyMol (Schrödinger, Inc., New York, NY, USA) and PDB ID 2HRC.

## Data Availability

Not applicable.

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
