# Peer review of "New Avenues of Heme Synthesis Regulation"

_ijms, 2022, doi:10.3390/ijms23137467_

Round 1

Reviewer 1 Report

This article by Medlock and Dailey reviewed novel aspects of the biochemistry of heme synthesis including the sources of glycine and succinyl- CoA as the precursors of tetrapyrrole macrocycle, mitochondrial heme metabolon and post-translational modifications such as phosphorylation, acylation, and glutathionylation of those rate-limiting enzymes. The role of immunometabolite itaconate  in heme biosynthesis and anemia of inflammation is also reviewed as well. This is a timely and useful review with also summary of recent findings and new data from the authors. An issue that could be further addressed  is on the relative dependence on the metabolic  rerouting in erythroid versus non- erythroid cells.

Fig. 7:

    Line 420: Define BioGEE

-                      Line 423: ….in some experiments,….

Author Response

Responses to Reviewer #1 are below in BOLD.

This article by Medlock and Dailey reviewed novel aspects of the biochemistry of heme synthesis including the sources of glycine and succinyl- CoA as the precursors of tetrapyrrole macrocycle, mitochondrial heme metabolon and post-translational modifications such as phosphorylation, acylation, and glutathionylation of those rate-limiting enzymes. The role of immunometabolite itaconate  in heme biosynthesis and anemia of inflammation is also reviewed as well. This is a timely and useful review with also summary of recent findings and new data from the authors. An issue that could be further addressed  is on the relative dependence on the metabolic  rerouting in erythroid versus non- erythroid cells.

We have added some text in the introduction to differentiate between erythroid and non-erythroid, but focused on erythroid heme synthesis and only in few instances refer to published data from nonerythroid heme synthesizing enzymes.

Fig. 7:

    Line 420: Define BioGEE

We have defined BioGEE as Biotinylated glutathione ethyl ester in the text in line 396.

-                      Line 423: ….in some experiments,….

The authors are unclear on what is suggested here.

Reviewer 2 Report

The subject presented in this review article is interesting: the mechanisms involved in the regulation of heme synthesis in erythropoiesis, with emphasis on post-translational control of enzymes through phosphorylation, succinylation, glutathiolynation and cofactors. The overall content is good, but certain sections of the review article are difficult to follow. The authors should  limit to relatively recent findings as other review articles are already available.  I think that the fact of addressing heme synthesis in non-erythroid and erythroid cells is confusing. It would be better to refer exclusively to heme synthesis in erythroblasts.

Specific comments.

11. The first paragraph (Lines 23-34) seems irrelevant for the purpose of the review.  In the second paragraph (line 35) erythropoiesis should be better explained, including the stages that are later mentioned, in which heme/hemoglobin synthesis is initiated. The pivotal role of GATA1 on induction of heme synthesis should be addressed.

22. It is unclear what is meant by the complement of enzymes (line 50). I guess it is meaning the 8 enzymes involved.  Groupe of enzymes would be more adequate.  

33.  Before comparing heme biosynthesis in non-erythroid and erythroid cells, it is advisable to briefly remind heme’s structure, as the authors refer to synthesis of tetrapyrrole macrocycle and protoheme (with no figure reminding the structure of heme) and authors should present the 8 reactions involved (in a figure).  As presented, the information is difficult to follow, specially for people unfamiliar with the subject (which are those that will be more interested in reading this review article).

44. The authors should remind that in non-erythroid cells, the first reaction is catalyzed by ALAS1, an enzyme controlled by intercellular heme levels, at the transcriptional and translational levels. 

55. Sentence in line 52 is difficult to understand: "Both housekeeping and erythroid-specific transcriptional elements exist within several genes encoding enzymes of heme synthesis… "exist should be replaced by …control several genes encoding.

66. The paragraph  in lines 50-63 is extremely dense, referring to location in chromosomes, GATA-1 action, and post-translational control of ALAS2. Too much information that is not well interconnected. This seems a summary of several things, that should be avoided. Information must be presented in a more organized manner. If the intention is to announce what will be  next discused, it should be done in a simpler manner.   If authors are willing to refer more specifically to organic precursors involved and PTM, unrelated information should not be presented. Rather, the readers should be directed to other reviews.     

87. Line 98 directs to one of the major subjects discussed. It is important to include a figure summarizing the structure of the different precursors leading to heme. Without a figure the text is difficult to understand.  

98. The simultaneous comparison between erythroid and non-erythroid cells is confusing. It would be better to refer first to non-erythroid cells. A second section could deal with erythroid cells (see general comment).

89. Figure 7 shows data that should be rather summarized in the text. As presented, the data are adequate for a research paper, not a review article.

 10. Dysfunctions and pathologies should be addressed at the last part of the review. There is no fluent connection between the different subjects adressed.

Author Response

Response to Reviewer #2 are below in BOLD.

The subject presented in this review article is interesting: the mechanisms involved in the regulation of heme synthesis in erythropoiesis, with emphasis on post-translational control of enzymes through phosphorylation, succinylation, glutathiolynation and cofactors. The overall content is good, but certain sections of the review article are difficult to follow. The authors should  limit to relatively recent findings as other review articles are already available.  I think that the fact of addressing heme synthesis in non-erythroid and erythroid cells is confusing. It would be better to refer exclusively to heme synthesis in erythroblasts.

The review is focused on erythroid heme synthesis and only in few instances do we refer to published data from nonerythroid heme synthesizing enzymes mainly to contrast important difference.

Specific comments.

  1. The first paragraph (Lines 23-34) seems irrelevant for the purpose of the review. In the second paragraph (line 35) erythropoiesis should be better explained, including the stages that are later mentioned, in which heme/hemoglobin synthesis is initiated. The pivotal role of GATA1 on induction of heme synthesis should be addressed.

The introductory paragraph is intended to put the complexity of erythroid heme synthesis in an evolutionary context. In the second paragraph we have added additional text related to  the role of GATA1. The stages during which heme is synthesized is already stated in the last sentence of paragraph 2. We would mention that there exist abundant publications concerning GATA1 and felt that review of gene regulation was outside the target of our current contribution which focuses on more downstream regulatory features.

However on original line 37 we have now added: “ This process which is initiated by the action of GATA1 starts with HSC…”

  1. It is unclear what is meant by the complement of enzymes (line 50). I guess it is meaning the 8 enzymes involved. Groupe of enzymes would be more adequate.

We have replaced “complement” with “All enzymes necessary…” and then cite new Figure 2 that shows all pathway intermediates and pathway enzymatic steps. 

  1. Before comparing heme biosynthesis in non-erythroid and erythroid cells, it is advisable to briefly remind heme’s structure, as the authors refer to synthesis of tetrapyrrole macrocycle and protoheme (with no figure reminding the structure of heme) and authors should present the 8 reactions involved (in a figure). As presented, the information is difficult to follow, specially for people unfamiliar with the subject (which are those that will be more interested in reading this review article).

Original Figure 3 shows the structure of protoporphyrin IX. However, we have added a new figure (Figure 2) that shows the pathway and structures of intermediates and cite this in the text.

  1. The authors should remind that in non-erythroid cells, the first reaction is catalyzed by ALAS1, an enzyme controlled by intercellular heme levels, at the transcriptional and translational levels.

We have added text to address this comment in the third paragraph.  Specifically we now state at original line 56: “ALAS1 expression is subject to regulation by a variety of factors that are beyond the scope of the present review (Medlock and Dailey, #10).,Then after the next sentence  that identifies ALAS2, we now state; “In both non-erythroid and erythroid precursor cells synthesizing heme, it is generally accepted that ALAS activity represents the rate limiting step of the pathway.”

  1. Sentence in line 52 is difficult to understand: "Both housekeeping and erythroid-specific transcriptional elements exist within several genes encoding enzymes of heme synthesis… "exist should be replaced by …control several genes encoding.

We have rewritten this sentence.

  1. The paragraph in lines 50-63 is extremely dense, referring to location in chromosomes, GATA-1 action, and post-translational control of ALAS2. Too much information that is not well interconnected. This seems a summary of several things, that should be avoided. Information must be presented in a more organized manner. If the intention is to announce what will be next discussed, it should be done in a simpler manner.   If authors are willing to refer more specifically to organic precursors involved and PTM, unrelated information should not be presented. Rather, the readers should be directed to other reviews.  

The reviewer’s point here and above (22 and 55) is taken. We have rewritten this paragraph as described above.   

  1. Line 98 directs to one of the major subjects discussed. It is important to include a figure summarizing the structure of the different precursors leading to heme. Without a figure the text is difficult to understand.

As mentioned above, we have added a new Figure 2 to respond to this and the earlier point.

  1. The simultaneous comparison between erythroid and non-erythroid cells is confusing. It would be better to refer first to non-erythroid cells. A second section could deal with erythroid cells (see general comment).

We are not clear about what the reviewer is requesting since we have focused on erythroid heme synthesis and only in few instances refer to published data from nonerythroid heme synthesizing enzymes.

  1. Figure 7 shows data that should be rather summarized in the text. As presented, the data are adequate for a research paper, not a review article.

We included the data figure since it more succinctly presents the information than explaining everything in text. It also allows readers to visually assess for themselves the nature of the proposal.

  1. Dysfunctions and pathologies should be addressed at the last part of the review. There is no fluent connection between the different subjects adressed.

Diseases and pathologies related to individual items discussed have been presented within the text, but we agree that a synopsis with associated references in the Conclusion section does improve the overall presentation. Thus, we have added a short summary and references at this point.